# Consumer preference to utilise a mobile health app: A stated preference experiment

**David Lim** *, **Richard Norman, Suzanne Robinson**

School of Public Health, Curtin University, Perth, Australia

* David.K.Lim@curtin.edu.au

## Abstract

### Background

One prominent barrier faced by healthcare consumers when accessing health services is a common requirement to complete repetitive, inefficient paper-based documentation at multiple registration sites. Digital innovation has a potential role to reduce the burden in this area, through the collection and sharing of data between healthcare providers. While there is growing evidence for digital innovations to potentially improve the effectiveness and efficiency of health systems, there is less information on the willingness of healthcare consumers to embrace and utilise technology to provide data.

### Aim

The study aims to improve understanding of consumers' preference for utilising a digital health administration mobile app.

### Methods

The online study used a stated preference experiment design to explore aspects of consumers' preference for a mobile health administration app and its impact on the likelihood of using the app. The survey was answered by a representative sample (by age and gender) of Australian adults, and sociodemographic factors were also recorded for analysis. Each participant answered eight choice sets in which a hypothetical app (defined by a set of dimensions and levels) was presented and the respondent was asked if they would be willing to provide data using that app. Analysis was conducted using bivariate logistic regression.

### Results

For the average respondent, the two most important dimensions were the time it took to register on the app and the electronic governance arrangements around their personal information. Willingness to use any app was found to differ based on respondent characteristics: people with higher education, and women, were relatively more willing to utilise the mobile health app.

**Data Availability Statement:** All three data files are available from the Research Data Australia database. Available at http://dx.doi.org/10.25917/5e256ca31c855.

**Funding:** The authors received no specific funding for this work.

**Competing interests:** The authors have declared that no competing interests exist.

## Conclusion

This study investigated consumers' willingness to utilise a digital health administration mobile app. The identification of key characteristics of more acceptable apps provide valuable insight and recommendations for developers of similar digital health administration technologies. This would increase the likelihood of achieving successful acceptance and utilisation by consumers. The results from this study provide evidence-based recommendations for future research and policy development, planning and implementation of digital health administration mobile applications in Australia.

## Introduction

There has been general acceptance by both consumers and physicians around the world that the current health system requires reformation to integrate with more technologically advanced means of exchanging information [1–4]. This is supported by a growing body of evidence relating to the potential utilisation of digital innovations to increase effectiveness and efficiency of health services and systems [5]. In Australia, a major impediment to progress is the fragmentation of its healthcare system, with healthcare provision and funding provided by multiple layers of government, the insurance sector, and by the individual themselves. Data generated at each layer are rarely shared between different funders, or between funders and payers. Thus, patients are often required to duplicate data when registering personal information, which is likely to be onerous and inefficient [6–8].

There is often frustration from consumers when they are required to provide similar (or identical) information at different points of the health system [6–8]. Additionally, many consumers are also faced with health illiteracy or language barriers that led to paperwork being completed inaccurately [9]. Healthcare providers also identified administrative documentation as a barrier towards provision of optimal patient care, and this is likely to be exacerbated by limited sharing of information from one health organisation to another [10, 11].

In Australia, as elsewhere, the government has recognised that self-recording of digital clinical data can provide much needed benefits to the health system. However, the country has been relatively slow in the uptake of digital innovation in health, including technologies such as electronic health records [2, 12, 13]. This was evident in 2009 when the Australian Government unsuccessfully launched the Person-Controlled Electronic Health Record (PCEHR), which was later rebranded as My Health Record (MHR) in 2015 [5]. Although there was substantial financial investment in the program, poor uptake by both medical practices and consumers occurred [14]. This can be attributed to a number of key challenges namely a lack of an acceptable governance approach, unanswered questions about the usability of the digital system, and only a subset of data collected was clinically relevant [5, 14]. However, while there is growing evidence relating to the potential of digital innovations, there is less information on the willingness of consumers to embrace and utilise digital health technologies [5]. Given uptake of such technology is crucial to success, it is important to understand what affects consumers' willingness, as well as their barriers and enablers towards utilising and engaging with such technologies.

This study aims to investigate consumers' willingness to utilise a digital patient administration mobile application in Australia. This should provide guidance on the same question in similar industrialised nations experiencing the same challenges. The objectives of the study are: to explore barriers and enablers and their impact on affect consumers' willingness to

utilise mobile health administration apps; to explore consumer views on governance and usage of their data; and to inform future planning and development of digital patient administration mobile apps. Evidence from this study can be used to inform future policy development, planning and implementation of digital health administration mobile apps in Australia, while broadening the knowledge on consumer views on governance and data management for digital health technologies.

## Methods

Stated preference approaches (including techniques such as discrete choice experiments (DCEs)) are increasingly being used in health because of their capability to quantitatively evaluate preferences [15]. Unlike revealed preference approaches, it can assess options that do not yet exist, and can more easily disentangle the effects of multiple factors on individual choice through appropriate experimental design. This allows for measurement and identification of the relative strength of preference for individual factors that combine to determine choice. Prior approval was obtained from the Curtin University Human Research Ethics Committee (HREC) after developing the survey instrument and before commencement of data simulations.

The research team began the experiment design process by identifying the dimensions and corresponding levels. Existing literature identified three overarching themes into which barriers and enablers of digital technology uptake were categorised. These three themes were broadly: service delivery to consumers [2, 16–21]; technology facilitation by staff members [2, 13, 22–27]; and strategic organisational factors [1–3, 18, 22, 24, 28–30]. The barriers and enablers identified were dependent on the purpose and objectives of each study. The type of studies included in the review either analysed factors within a single category, or across more than one category. The team subsequently selected nine different dimensions that were of interest for the purposes of the study, each dimension having four possible levels. A full factorial design (consisting of every possible combination of levels) would have generated 262,144 ($4^9$) different options. Instead, an orthogonal array was implemented to arrange each corresponding level from every dimension into specific combinations. There were a total of 32 choice sets, with all dimensions and leach level appearing an equal number of times—as illustrated in S1 Appendix. Each participant was presented with eight choice sets selected at random from the 32. Only 8 choice sets per asked per respondent to improve the response rate and not to overburden participants. Fig 1 details all dimensions and levels that were allocated according to the experiment design from S1 Appendix.

The experiment was administered online and was facilitated by SurveyEngine, a company specialising in the administration of online surveys. Potential respondents were drawn from an online panel of general population individuals, who have stated their willingness to participate in research. They were invited to the survey via a weblink and given the option to participate. If they were willing to do so and were within the sampling frame, they then received an introduction to the task, and completed the eight choice sets. Finally, the survey collected feedback about their experience of the survey along with additional covariate sociodemographic information from participants. Specifically, participants reported their geographic location, primary language, level of education, history of chronic conditions, and level of income. The full survey instrument is provided in S2 Appendix.

### Sampling frame

The survey was administered in a sample of 500 Australian adults, representative of the general population in terms of age and gender defined by the Australian Bureau of Statistics (ABS)

| Dimensions | | Levels | |
|---|---|---|---|
| **No.** | **Description** | **No.** | **Description** |
| **1** | **Registration Time** Complete registration on the app will take approximately: | 0 | 5 mins |
| | | 1 | 10 mins |
| | | 2 | 20 mins |
| | | 3 | 30 mins |
| **2** | **Waiting Time** It will reduce your waiting time for subsequent visits at the medical centre by approximately: | 0 | 0 mins |
| | | 1 | 5 mins |
| | | 2 | 10 mins |
| | | 3 | 20 mins |
| **3** | **Type of information** It will be required of you to provide information on: | 0 | Your personal and family medical history and your medications |
| | | 1 | Your personal and family medical history, your medications and the number of tobacco products you smoke |
| | | 2 | Your personal and family medical history, your medications, the number of tobacco products you smoke and the amount of alcohol you drink |
| | | 3 | Your personal and family medical history, your medications, the number of tobacco products you smoke, the amount of alcohol you drink and the use of illicit drugs |
| **4** | **Privacy** Clinically relevant information that you provided on the app will be available to: | 0 | Your treating doctor attending to you |
| | | 1 | Your treating doctor attending to you and other allied health professionals (ie. pharmacists, physiotherapists, clinicians and nurses) |
| | | 2 | Your treating doctor attending to you, other doctors, allied health professionals and the Australian Government |
| | | 3 | Your treating doctor attending to you, other doctors, allied health professionals, the Australian Government and insurance companies |
| **5** | **Governance** The storage and use of the information you provided will be guided with policies and regulations implemented by: | 0 | The medical centre |
| | | 1 | The Australian Government |
| | | 2 | A private consultancy firm |
| | | 3 | None |
| **6** | **Support** Support staff will be available to help you register on the app through: | 0 | Face-to-face at the medical centre |
| | | 1 | Over the phone |
| | | 2 | E-mail |
| | | 3 | No support available |
| **7** | **Convenience** Upon complete registration on the app, it will give you the option to: | 0 | Book an appointment with your desired health professional |
| | | 1 | Reschedule appointments in advance |
| | | 2 | Send notification reminders about your next appointment on your smart phone |
| | | 3 | Does not offer any assistance with booking appointments |
| **8** | **Research** The data you provided will be anonymized and may be used for research by: | 0 | Local universities |
| | | 1 | Government researchers |
| | | 2 | Private pharmaceutical companies |
| | | 3 | No one |
| **9** | **Risk** Upon complete registration, it **reduces** your risk of a medical error by: | 0 | 0% |
| | | 1 | 1% |
| | | 2 | 5% |
| | | 3 | 10% |

**Fig 1. Dimensions and levels.**

[31]. A recent review of the field suggested that a sample size of 500 was typical for these kinds of studies (which had a median of 401) [32]. There were also screening questions that only accepted respondents who own a smartphone and attend an appointment at a medical centre annually. Thus, filtering out those who were likely to be non-users of the technology for reasons other than preference.

Analysis was conducted using logistic regression in STATA, with standard errors adjusted to reflect the clustering of responses within each respondent. First, the entire sample was used to generate the mean preferences for the dimensions and levels of interest in the study. Second, the sample was analysed against sociodemographic factors to examine if different characteristics of respondents impacts on their willingness to utilise the app. Additionally, to examine if different aspects of the app were relatively more important for different kinds of people. These characteristics were analysed as subgroups listed as the following: gender, age ($<$55 years vs 55 +), geography (metropolitan vs non-metropolitan), whether the individual has a chronic condition, education (diploma or higher vs lower qualifications), Aboriginal and Torres Strait Islander status, and primary language (English vs non-English). Analysis was performed using separate regressions for each subgroup.

## Results

In total, 511 participants successfully completed the survey. Table 1 compares survey completers with the Australian adult population.

Additionally, there were 36 incomplete surveys who did provide a response to at least one of the stated preference choice sets. While the demographic information for these respondents were incomplete, they were included in the analysis set, yielding a total of 547 respondents. Fig 2 presents the final overall bivariate analysis and also bivariate analysis for each subgroup.

### Analysis and evaluation of final data

Consumers' willingness to utilise the mobile app was most associated with two broad areas identified in the study. Specifically they relate to the time it takes to complete the registration on the mobile health app, and insecurities regarding management of their information. Registration time was the highest significant ($p<0.01$) dimension that deterred the participants' willingness to utilise the mobile app (see Fig 2).

The other dimension of interest that relates to data insecurity is 'Governance'. Respondents demonstrated a strong preference for either governmental or medical centre governance over no governance structure or a structure defined by a private consultancy firm. Furthermore,

**Table 1. Comparison between sampling frame vs. completed participants.**

| Age group (years) | Sampling frame | | | Completed participant quota (n = 511) | | |
| --- | --- | --- | --- | --- | --- | --- |
| | Percentage (%) out of all Australians* | Percentage (%) of adult Australians | Gender ratio as at December 2017 from the ABS Male : Female | Male | Female | Percentage (%) |
| 18–24 | 9.49 | 12.23 | 1.05 | 30 | 32 | 12.13 |
| 25–54 | 41.18 | 53.05 | 0.98 | 135 | 133 | 52.45 |
| 55–64 | 11.54 | 14.87 | 0.96 | 37 | 43 | 15.66 |
| 65 and over | 15.41 | 19.85 | 0.88 | 50 | 51 | 19.77 |
| Subtotal: | 77.62 | 100 | | | | |

*Total population of Australians from all ages was 24,597,528 [31].

**Fig 2. Data analysis and evaluation.**

this concern for data security was supported in the results for the dimension of 'Research'. Where providing information for research to private pharmaceutical companies also displayed a significant negative impact on respondents' willingness to utilise the app (p<0.05).

There was no significant association between the population's willingness to utilise the app and allowing insurance companies access personal information. A similar level of association was seen with Government researchers given access to personal information. There was also no significant impact on the type of information people were willing to provide, especially with information about illicit drug use and more detailed personal and family history.

The dimension of 'Support' was generally not well received by the public. Support via over-the-phone and email both had significant negative associations. Additionally, there was no statistically significant association shown overall for having no support relative to the base of face-to-face support. Additionally, there was also no significant positive association between people's willingness and the dimensions for convenience and reduction in risk of medical errors.

## Subgroup analysis

By examining the constant values and the corresponding level of significance in Fig 2, it reveals an order of likelihood for subgroups within the population to utilise the mobile health app. Table 2 lists the order of likelihood, starting from 1 being most willing.

**Table 2. Subgroup order of likelihood to utilise mobile health app.**

| | Subgroup |
|---|---|
| 1 | Higher education level (Diploma level and above) |
| 2 | No long-term medical conditions |
| 3 | Young (aged between 18–54) |
| 4 | High income earner ($84,000 and above) |
| 5 | Lives in the metropolitan region |
| 6 | Not Aboriginal or Torres Strait Islander |
| 7 | Primary language other than English |
| 8 | Male |

The first subgroup being the most willing to utilise the app were those with a higher education level (p<0.01). However, this group had a high negative association (p<0.01) with governance for the mobile app implemented by a private consultancy firm, or when there is no governance structure at all. Moreover, they also displayed significant negative association with over-the-phone support.

The second most willing subgroup to utilise the app were those without chronic medical conditions (p<0.05). This group's willingness to utilise the app significantly increased with a reduction in the risk of medical errors. Whereas people with chronic medical conditions were not show any significant impact by reduction in risk of medical errors.

The third most willing subgroup were younger people (aged 18–54) (p<0.05), however they showed significant negative association with support over the phone or email. This highlights the need for improved communication between patients and healthcare providers, otherwise it will continue to be a significant barrier towards uptake of digital technology [4]. Furthermore, this group was significantly positively influenced by having a reduction in risk of a medical error. In comparison to younger people, older people were less negatively impacted by the registration time to utilise the mobile app. However, older people were to a greater extent negatively associated with having no governance (p<0.01) or having a private consultancy firm (p<0.05) implement policies.

When comparing results by gender, no significant association was found between males and the impact of privacy issues or the type of support made available to help with registration on the app. Contrastingly, females were very negatively associated with all forms of support. The same sentiment was shown towards the Australian Government or insurance companies having access to personal information for females.

Lower income earners were negatively associated with significance across dimensions such as privacy, governance and support. This agrees with broader literature that examines how low socioeconomic status poses as a significant barrier towards digital uptake [16, 20, 21]. Unlike low income earners, higher income earners were not significantly negatively affected by most of the levels and resulted in having a high positive constant value of 0.57 with low significance. This meant that high income earners were more willing to utilise the mobile app when compared to low income earners.

Having information made available to their doctor and allied health professionals showed a high significant positive association for people living in rural regions. This demonstrates an ongoing problem faced by those living rurally being inaccessible to healthcare due to their geographical location [33, 34]. This finding may indicate that perhaps by having their medical information made available to healthcare providers on the mobile app, it may help bridge this gap. Conversely, people living in metropolitan areas were not impacted by any level of privacy, but were positively influenced by having a risk reduction. Moreover, the metropolitan group was significantly more likely to utilise the mobile app than those living rurally.

Aboriginal or Torres Strait Islanders showed very high significance for almost all levels either positively or negatively. However, this result is likely to be inaccurate due to very small numbers in this subgroup (n = 15).

## Discussion

To our knowledge, this study is the first quantitative study in Australia to explore consumers' perceptions on their utilisation of a digital health mobile app. Results demonstrated registration time and governance structure were important for respondents. Respondents were strongly opposed to spending time registering on the mobile app in order to use it. Given the implications of registration time on uptake and usage, app developers need to consider how to

provide a seamless registration process that can eliminate lengthy registration time. Conversely, certain dimensions, such as risk reduction and reduction in waiting time did not necessarily translate into a greater willingness to provide data through the app. Respondents appeared willing to provide information on dimensions that were potential barriers, such as providing information on usage of illicit drugs, or sharing information with the Australian Government or insurance companies [35, 36].

While the results from this study do echo some of the existing literature, it is notable that there are a number of clear points of difference between our work and others. For example, an Australian study suggested that waiting times to see the doctor was of significant concern [37]. However, our results showed that a reduction in waiting time was only a statistically significant factor for a small number of sub-groups (specifically people with lower education level and those living rurally). This could be due to the time interval stated on the levels of the 'waiting time' dimension was not large enough for participants to deem as significant for them. This was reflected in some respondents' comments who wanted to see larger reduction in waiting time to choose from, which would provide more significant benefits from a consumer's perspective.

People whose primary language was not English showed a very significant positive association with the several dimensions. Namely around convenience, type of information and data being included in research. This could be attributed to the app presenting itself as a potential solution capable of overcoming the language barrier [4, 38, 39]. Potentially the app could have a list of different languages to choose from, that would allow a diverse range of people to utilise the app with complete understanding.

When respondents whose primary language was English were examined, they had a significant negative association towards having their information available for research to private pharmaceutical companies, governance by a private consultancy firm, or no governance. This emphasises an underlying concern about the security and privacy of their information. This mirrors a general finding from literature of a poor understanding on how consumers' data is managed, mostly unregulated and offers no protection for consumers for their digital health data [40, 41].

Findings from our study reinforced an underlying apprehension by participants around data governance and usage of their data. The results showed negative associations with information shared with either insurance companies or the Australian Government. There was also strong negative associations for both governance by private consultancy firms or no governance, and for private pharmaceutical companies using consumers' information for research. Overall, these negative associations conveyed a withdrawal of participants' willingness to use the app when their personal information reaches the public domain, or when the intention to use of their data is unclear. The results from this study suggest that individuals are more willing to share sensitive information, if the use of their data was to support research activity and or could have the potential to reduce medical errors [22, 26, 42].

Going forward, more detail on how information is being used for research needs to be communicated with users and potential users. Suggestions for engagement include more communication with consumers in relation to the structure and development of the app. Such engagement could improve consumers' acceptance and willingness to utilise the app.

There are a number of study limitations that should be acknowledged. The target population only included people with a moderate to high level of financial capacity and technological literacy; being able to afford and use a smartphone and a computer. Whilst the study is not representative of the entire Australian adult population, a recent review undertaken in 2015, noted that 15.3 million Australians have access to a smartphone and 11.2 million have access to a tablet, demonstrating the extensive coverage of mobile technology across the Australian population [43].

Another limitation related to the methods used in the stated preference experiment which applied predominately quantitative data analysis. Future studies could do more to combine quantitative and qualitative questions, this would enrich the data collection and add meaning to quantitative results [44]. For example, future research could focus on investigating possible associations that relate to: the impact of psychological factors such consumers' motivational level and their perceived level of health literacy. In addition, other barriers and enablers related to technology that are outside of an individual consumer's control, including those relating to health professionals or other organisational factors could also be of interest.

## Conclusion

This study investigated key factors on consumers' preference towards utilisation of a digital administration mobile app. The two most significant dimensions were the time it took to register on the app and data governance structure for the app—with greatest concern on electronic management of personal information provided on the app. This underlines crucial aspects from the broader literature for a need for an improved public understanding towards data security and transparency of consumers' online data. Although there are significant barriers to the uptake of such digital health technologies, there are potential areas of growth that could be further developed. Especially in areas such as its collection of data for research, reduction in medical errors, bridging the language barrier for all users, and potentially improving the accessibility of healthcare services for those living rurally.

Future research should continue to investigate into solutions for the barriers surrounding the uptake of digital health technologies, and policy makers should address these issues adequately. This would provide a more comprehensive knowledge on the uptake of digital health administration tools and assist with transition of healthcare systems to become completely digital. Ultimately, this would enhance patient outcomes and improve the patient journey along healthcare systems.

## Supporting information

**S1 Appendix. Orthogonal array- 9 dimensions with 4 levels.**
(PDF)

**S2 Appendix. Page-by-page online survey.**
(PDF)

## Author Contributions

**Conceptualization:** David Lim.

**Data curation:** David Lim.

**Formal analysis:** David Lim.

**Investigation:** David Lim.

**Methodology:** David Lim.

**Project administration:** David Lim.

**Supervision:** Richard Norman, Suzanne Robinson.

**Writing – original draft:** David Lim.

**Writing – review & editing:** David Lim.

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
