## [Decision Letter · Decision Letter 0]

12 Dec 2019

PONE-D-19-28394

Consumer preference for a digital health administration mobile app: A stated preference experiment

PLOS ONE

Dear Mr Lim,

Thank you for submitting your manuscript to PLOS ONE. After careful consideration, we feel that it has merit but does not fully meet PLOS ONE’s publication criteria as it currently stands. Therefore, we invite you to submit a revised version of the manuscript that addresses the points raised during the review process.

We would appreciate receiving your revised manuscript by 11th January 2020. To enhance the reproducibility of your results, we recommend that if applicable you deposit your laboratory protocols in protocols.io, where a protocol can be assigned its own identifier (DOI) such that it can be cited independently in the future. For instructions see: http://journals.plos.org/plosone/s/submission-guidelines#loc-laboratory-protocols

We look forward to receiving your revised manuscript.

Kind regards,

Kwasi Torpey, MD PhD MPH

Academic Editor

PLOS ONE

Journal Requirements:

1. We note that you have indicated that data from this study are available upon request. PLOS only allows data to be available upon request if there are legal or ethical restrictions on sharing data publicly. For information on unacceptable data access restrictions, please see http://journals.plos.org/plosone/s/data-availability#loc-unacceptable-data-access-restrictions.

Additional Editor Comments (if provided):

The manuscript titled " Consumer preference for a digital health administration mobile app: A stated preference experiment" seeks to investigate consumer willingness to use digital patient administration technology in Australia. Though the study is interesting there are a number of key issues that needs to be addressed

1. What was the basis of the sample size of 500?

2. What approach was used to ensure adequate representation by age and gender- These are poorly described under the methods section

3. The manuscript needs thorough copyediting paying attention to "long sentences" and inappropriate punctuations particularly commas

Reviewers' comments:

Reviewer's Responses to Questions

**Comments to the Author**

1. Is the manuscript technically sound, and do the data support the conclusions?

Reviewer #1: Yes

2. Has the statistical analysis been performed appropriately and rigorously? 

Reviewer #1: No

3. Have the authors made all data underlying the findings in their manuscript fully available?

Reviewer #1: No

4. Is the manuscript presented in an intelligible fashion and written in standard English?

Reviewer #1: Yes

5. Review Comments to the Author

Reviewer #1: Objectives

1. Authors have stated the following objectives of the study:

- To explore the barriers and enablers that affect consumers’ willingness to utilise mobile health administration apps

- To explore consumer views on governance and usage of their data

- To inform future planning and development of digital patient administration mobile applications.

Authors should clarify if this manuscript describes a portion of a larger study. It is unclear if these stated objectives were to be achieved in this current manuscript.

2. The third stated objective is not achievable as an objective of this paper.

Methods

1. Authors mention conducting a literature review in identifying dimensions and corresponding levels. More detail should be provided on the type of review conducted and how it was conducted.

2. What are the dimensions and corresponding levels being referred to in the paper? Authors are vague in their description of these concepts.

a. What is / are dimensions?

b. What is/are corresponding levels?

c. What is/ are choice sets? What makes up a choice set?

3. The authors should be specific on what online panel the study participants were drawn from. How exactly were they identified and invited using the weblink?

4. How was the sample of 500 respondents arrived at?

5. Was the logistic regression bivariate or multivariable?

Results

1. What are the barriers and enablers that affect consumers’ willingness to utilise mobile health administration apps? Findings to this research questions should be stated more explicitly.

2. Were the logistic regression results presented as a result of multivariable analysis? Authors should specify.

General comments

Authors should submit the manuscript for professional copy editing. Some sentences are too long and incoherent.

Title of the manuscript should be reviewed to reflect the content more closely. E.g. Consumer willingness to provide data through mobile app: A stated preference experiment

6. PLOS authors have the option to publish the peer review history of their article (what does this mean?). If published, this will include your full peer review and any attached files.

Reviewer #1: No

---

## [Author Response · Author response to Decision Letter 0]

31 Jan 2020

1. What was the basis of the sample size of 500?

A recent review of the field suggested that a sample size of 500 was typical for these kinds of studies (which had a median of 401) [1]. The Australian Bureau of Statistics website also provides a Sample Size Calculator that recommends a sample size of only 385 for a population size of 24,597,528 with a p value of 0.05. (https://www.abs.gov.au/websitedbs/d3310114.nsf/home/sample+size+calculator)

2. What approach was used to ensure adequate representation by age and gender?

Participants were recruited through SurveyEngine, who have confirmed they used Toluna Australia (an online panel of Australian residents). The online survey (see S2 Appendix) had filtering questions applied to representative percentages of age (18-24, 25-54, 55-64, and 65 and older) and gender according to the Australian Bureau of Statistics (ABS). Participants were invited to the survey via link and given the option to participate. If they were willing to do so, they would then be required to provide information to ensure they meet the eligibility criteria to qualify as part of the representative sample of the population. Assuming the quota was not met yet, they then continue to complete the main survey.

3. The manuscript needs thorough copyediting paying attention to "long sentences" and inappropriate punctuations particularly commas.

Please see updated manuscript and also the manuscript with track changes. Editing has been performed with close attention to long sentences and use of appropriate punctuation.

4. Reviewer #1: Objectives

Authors have stated the following objectives of the study:

- To explore the barriers and enablers that affect consumers’ willingness to utilise mobile health administration apps

- To explore consumer views on governance and usage of their data

- To inform future planning and development of digital patient administration mobile applications.

Authors should clarify if this manuscript describes a portion of a larger study. It is unclear if these stated objectives were to be achieved in this current manuscript.

The manuscript is not part of a larger study, but rather it was to contribute towards additional knowledge.

5. The third stated objective is not achievable as an objective of this paper.

Third stated objective has been changed to now read: To provide evidence-based recommendations for future research and development of digital health administration mobile applications.

Methods

6. Authors mention conducting a literature review in identifying dimensions and corresponding levels. More detail should be provided on the type of review conducted and how it was conducted.

We thank the reviewer for this comment. We did not provide significant detail about the review due to space issues in the manuscript, but have prepared the following text which could be included if the editor is happy for the manuscript to be a little longer to allow this.

“The review was conducted in ProQuest and Medline, with the intention of summarising and examining relevant knowledge on the topic. The chosen search strategy is been outlined below, with keywords chosen based on a scoping search of the literature. The location was set to Australia as the pilot study for the digital administration tool was specifically being implemented in this country. However, Medline database did not have the filter option for Australia and included publications from all countries. This provided a wider review of digital technologies that were studied in other country’s health systems and served to improve this literature review’s credibility.

Table 1: Initial literature review on ProQuest using keywords

Keywords in advanced search Applied filters

"digital readiness*" OR 

"digital technology" 

AND general practice" OR "consumer*" OR patient* Location: Australia Peer reviewed articles 

 Date range:

2008-2018

The initial search on ProQuest identified 40 potentially relevant studies, while the Medline search provided 179 results using a similar set of key terms and criteria. The search was then further narrowed down by adding more search terms as shown in Table 2.

With the additional keywords, the search on ProQuest was narrowed down to 24 results and Medline gave 22 results. The articles were first screened for relevance, according to their titles and abstracts, excluding any duplicates and irrelevant articles. 

The advanced search included the following:

("digital readiness*" OR "digital technology") AND ("general practice" OR "consumer*" OR patient*) AND ("ad?pt*" OR uptake OR usage OR barrier* OR enabler*) AND ("methods")

Figure 1: Flow diagram of literature review. From Moher D, Liberati A, Tetzlaff J, Altman DG, The PRISMA Group (2009). Preferred Reporting Items for Systematic Reviews and Meta-Analyses: The PRISMA Statement. PLoS Med 6(7): e1000097. doi:10.1371/journal.pmed1000097

Figure 1 illustrates the search strategy from both databases provided a combined total of 46 results. Additional manual hand searching using relevant keywords yielded 4 more results. The results were pre-screened for any duplicate entries and found 1 duplicate result to be removed. Entries were hand screened to ensure that articles only included peer reviewed publications dated between 2008 to 2018 inclusively, to allow for relevant and contemporary information to be analysed; since information technology advances at a fast pace. All results were then initially screened by their titles and abstracts, and publications that did not involve relevant adoption of a digital technology platform among consumers or organisation’s staff in the health sector were also excluded. Following on, 45 full-text articles were examined for their suitability and appropriateness in analysing barriers and enablers among the three categories. A remainder of 29 full-text articles were considered eligible and were examined for the objectives of this review.

The literature review identified three overarching themes where barriers and enablers for the uptake of digital technology were categorised. These three themes were broadly: service delivery to consumers; technology facilitation by staff members; and strategic organisational factors. The barriers and enablers identified in each study were dependent on the purpose and objectives of the study. The type of studies included in the review either, analysed factors within a single theme, or across more than one theme.”

7. What are the dimensions and corresponding levels being referred to in the paper? Authors are vague in their description of these concepts.

Thank you for the comment. We acknowledge the value of explaining more fully the use of these terms in our context. 

a. What is/are dimensions?

Each dimension in this study refers to an area or theme which may impact on a person’s decision to complete the data collection tool. These were identified in the literature review, and are listed in Figure 1 in S1 Appendix.

b. What is/are corresponding levels?

Each level is a possible value that each of the dimensions can take. These are listed in Figure 1 in S1 Appendix.

c. What is/ are choice sets? What makes up a choice set?

An example of a choice set is given on page 9 of S2 Appendix. It is the given task for the participant to complete, as shown in the attached survey in S2 Appendix.

8. The authors should be specific on what online panel the study participants were drawn from. How exactly were they identified and invited using the weblink?

The experiment was administered online, facilitated by SurveyEngine. This is a company which specialises in the administration of this kind of experiment. SurveyEngine has advised that the online panel that they used was Toluna Australia to conduct the study. Participants were drawn from the Australian population and filtered by their age and gender.

9. Was the logistic regression bivariate or multivariable?

It was bivariate reflecting that the individual either agreed or disagreed to completing the hypothetical data collection app.

Results

11. What are the barriers and enablers that affect consumers’ willingness to utilise mobile health administration apps? Findings to this research questions should be stated more explicitly.

The text has been edited to better reflect the objective of the study, “…to explore barriers and enablers and their impact on consumers’ willingness to utilise mobile health administration apps…”

10. Were the logistic regression results presented as a result of multivariable analysis? Authors should specify.

Text in results are edited to say “Figure 2 presents the final overall bivariate analysis and also bivariate analysis for each subgroup.”

11. Title of the manuscript should be reviewed to reflect the content more closely. 

Title has been updated to: Consumer preference to utilise a mobile health app: A stated preference experiment

 

References

1. Soekhai V, Bekker-Grob E, Ellis A, Vass C. Discrete Choice Experiments in Health Economics: Past, Present and Future. PharmacoEconomics. 2019;37(2):201-26. doi: 10.1007/s40273-018-0734-2.

---

## [Editor Report · Decision Letter 1]

10 Feb 2020

Consumer preference to utilise a mobile health app: A stated preference experiment

PONE-D-19-28394R1

Dear Mr Lim,

We are pleased to inform you that your manuscript has been judged scientifically suitable for publication and will be formally accepted for publication once it complies with all outstanding technical requirements.

With kind regards,

Kwasi Torpey, MD PhD MPH

Academic Editor

PLOS ONE
---

## [Editor Report · Acceptance letter]

12 Feb 2020

PONE-D-19-28394R1 

Consumer preference to utilise a mobile health app: A stated preference experiment 

Dear Dr. Lim:

I am pleased to inform you that your manuscript has been deemed suitable for publication in PLOS ONE. Congratulations! Your manuscript is now with our production department. 

With kind regards,

on behalf of

Professor Kwasi Torpey 

Academic Editor

PLOS ONE